# Effect of Abiotic Stresses from Drought, Temperature, and Density on Germination and Seedling Growth of Barley (*Hordeum vulgare* L.)

**DOI:** 10.3390/plants12091792

**Published:** 2023-04-27

**Authors:** Ákos Tarnawa, Zoltán Kende, Asma Haj Sghaier, Gergő Péter Kovács, Csaba Gyuricza, Hussein Khaeim

**Affiliations:** 1Institute of Agronomy, Hungarian University of Agriculture and Life Sciences, Páter Károly u.1, Gödöllő, 2100 Budapest, Hungary; 2Field Crop Department, College of Agriculture, University of Al-Qadisiyah, Al Diwaniyah 58002, Iraq

**Keywords:** abiotic stresses, drought, germination time, *Hordeum vulgare* L., seed germination, seedling growth, temperature

## Abstract

Seed germination and seedling growth are highly sensitive to deficit moisture and temperature stress. This study was designed to investigate barley (*Hordeum vulgare* L.) seeds’ germination and seedling growth under conditions of abiotic stresses. Constant temperature levels of 5, 10, 15, 20, 25, 30, and 35 °C were used for the germination test. Drought and waterlogging stresses using 30 different water levels were examined using two methods: either based at 1 milliliter intervals or, on the other hand, as percentages of thousand kernel weight (TKW). Seedling density in a petri dish and antifungal application techniques were also investigated. Temperature significantly impacted germination time and seedling development with an ideal range of 15–20 °C, with a more comprehensive range to 10 °C. Higher temperatures reversely affected germination percentage, and the lower ones affected the germination and seedling growth rate. Germination commenced at 130% water of the TKW, and the ideal water range for seedling development was greater and more extensive than the range for germination, which means there is a difference between the starting point for germination and the seedling development. Seed size define germination water requirements and provides an objective and more precise basis suggesting an optimal range supply of 720% and 1080% of TKW for barley seedling development. A total of 10 seeds per 9 cm petri dish may be preferable over greater densities. The techniques of priming seeds with an antifungal solution (Bordóilé or Hypo) or antifungal application at even 5 ppm in the media significantly prevented fungal growth. This study is novel regarding the levels and types of abiotic stresses, the crop, the experimental and measurement techniques, and in comparison to the previous studies.

## 1. Introduction

Barley (*Hordeum vulgare* L.) is one of the significant staple cereal crops that ranks fourth in terms of quantities produced and area cultivated behind maize, wheat, and rice [1]. Seemingly, it was primarily used as human food but evolved mainly into a feed, malting, and brewing grain, owing partly to wheat and rice’s ascent to importance [2]. Barley is cultivated and produced in wide climates, ranging from subarctic to subtropical areas [3]. As a result, this grain crop is critical in regions where food security is a concern [4,5]. Barley is a C3 crop species belonging to the Gramineae family and is moderately sensitive to abiotic stresses [6]. This means that barley is less sensitive to abiotic stresses than wheat; therefore, it can be grown in less favorable conditions. However, due to its global distribution and cultivation in various habitats, barley is subjected to various environmental abiotic stressors, including drought and increasing temperatures due to climate change, all of which result in severe yield losses [7,8]. Therefore, plants live in predominantly inappropriate or stressful environments for their development [9]. Abiotic stresses, such as severe climatic conditions (e.g., drought, flood, cold, heat, etc.), severely restrict crop development and yield formation [10]. These stress conditions may reduce seed germination while they can perform adaptation to induced abiotic stress [11].

Germination is essential in the reproduction and propagation of flowering plants [12,13]. It is a physiological stage that originates and develops a seedling via activating a series of biological and biochemical processes [14]. This physiological process commences when seeds absorb water rapidly, expanding and softening the seed coat and degrading the endosperm cell walls at the appropriate temperature [15]. Meanwhile, enzymatic digestion of carbohydrates and proteins is initiated to utilize nutrients for the budding plant [16,17]. Next, the seed’s internal physiological functions are triggered in association with the commencement of the seed’s respiration [18,19]. Ultimately, the ruptured seed coat enables the radicle and shoot to initiate [6,20,21]. Thus, it begins with the dormant dry seed absorbing water and is completed by the elongating of the embryo axis and, eventually, the emergence of the radicle [11,22,23]. Germination entails successive coordinated morphogenetic processes, such as energy relocation, endospermic nutrient absorption, and physiological and metabolic changes [24,25].

Germination impacts the quantity of barley’s definitive production and quality [26,27,28,29]. Environmental elements’ interaction with the seeds’ physiological state determines germination [30,31,32]. The demand for the different environmental abiotic factors is determined by the genotype’s reaction to these surrounding abiotic elements and these abiotic elements as a collective [33,34,35]. The internal seed’s response to overlapping external abiotic variables defines the success of the propagation [36].

Prosperous production necessitates the growth of robust, sophisticated seedlings, and one of the primary objectives of seedling production is to generate a viable plant from all seeds [37,38]. Endosperm materials’ availability, accessibility, and mobility are decisively critical in seedling establishment [39,40]. Temperature and water availability substantially impact the biochemical enzymic activity throughout these phases [41,42]. Plants cannot move freely under different abiotic stresses as they are sessile living organisms. As a result, they should respond to various unfavorable environmental conditions via many physiological and biological reactions to a specific limit [9].

The response to any kind of stress–even biotic or abiotic–can trigger a highly complex system [43]. In most cases, different stress factors have a measurable effect on changes in enzyme balance in barley [44], just as in other plants [45]. Therefore, the level of enzyme production and the pattern of its changes can be the base for handling the adverse effects. The possibilities of stress treatment are intensively researched and, for us, agrotechnical solutions [46] or breeding methods, even through the crossing of closer or more distant species [47], arise as an analogy for further work.

Temperature is an indispensable factor for germination, affecting all the different responses and phases of seed germination [15,48]. As temperature increases, the cell’s energy states and enzyme activity alter, and ATP content increases, but protein synthesis decreases [48,49]. Temperature stress boosts the transcription, translation, and activity of ROS scavenging enzymes, which leads to the formation and accumulation of H_2_O_2_ [50,51]. Germination is an intricate process for each phase-specific cardinal temperature scale, and the ideal temperature stimulates germination in the shortest time possible [13]. Sugar metabolism and respiration are primarily affected by any increase or decrease in the temperature, and the homeostasis of reactive oxygen species (ROS) requisite for germination is obstructed via anomalous respiration [15,52,53]. The temperature substantially affects germination duration, defined as the time interval between exposing the seed to water and commencing germination [11,54].

Water is required for seed germination to occur via hydrating the protoplasmic processes, provisioning dissolved oxygen, softening the hard outer coat of the seed, and ameliorating the permeability [15,55,56,57,58]. Water is necessary for the success of the metabolic processes of germination stages [59]. Its stress causes descending enzymic reactions, impacts carbohydrate metabolisms, lessens the water potential and the quantity of soluble calcium and potassium, and causes an alternation in the hormones [23,60,61].

As a ramification of continuing global warming and climate changes, it is foreseeable that temperature and drought will impact approximately half of all arable lands by the year 2050, while each of these abiotic stress can cause a remarkable yield reduction [62,63,64].

Seed germination and seedling growth pattern experiments under abiotic stress conditions have been performed for many years and are a common tool in plant research. However, research is lacking regarding the specific abiotic stress conditions being tested. This study is novel since it investigates several abiotic stress conditions (multiple levels of temperature, different bases of water application, different techniques of seed and media sterilization, and different seed and seedling densities, which is very useful for plant breeders, especially in the case of seed limitations), due to the plant species being studied (barley in this experiment), and due to the methods and techniques used to measure germination and growth, optimizing them and comparing them to previous studies.

### 1.1. Specific Aims of the Study

This study aims to scrutinize the germination capacity of barley seeds with various temperatures and water conditions and the proper densities of seeds and seedlings in a petri dish (PD) along with the antifungal inhibition technique. It must be underlined that these conditions refer directly to the experimental laboratory environment, as our prime aim was to clarify experimental methodology in the first step. Understanding the fundamentals of germination is critical in the realm of crop production. The current results are further confronted with information from the previous literature to refine germination and seedling establishment optima. The current research objectives are as follows: (i) To examine the impact of temperature on germination ability and duration, and the development of barley seedlings at six distinct temperature levels. (ii) To identify the optimal water limit for germination on two distinguished scales: at single-milliliter intervals and as percentages of thousand kernels weight (TKW). This technique was utilized to test the null hypothesis: “the divergence in the size and weight of the seeds does not affect the germination process when water quantity remains constant”. Hence, one objective is to evaluate the validation of the alternative hypothesis: “for germination, seed size and weight affect the quantity of water required”. Larger seeds, in theory, require more water to germinate than smaller ones. If the alternate hypothesis is valid, it will deliver a more nuanced approach for defining the ideal quantity of water necessitated for germination. (iii) To examine the impact of the number of seeds and density of seedlings in a PD on germination rate and on the open lid of a PD. This is fundamental for further scientific laboratory work. An open petri dish resulted in several negative sequelae, inclusive of contamination exposure and moisture loss. (iv) To investigate the effect on seedling growth of using the seed priming technique with antifungal or applying it to the growth media. This study was carried out following the Ministry of Agriculture and Rural Development’s No. 48 of 2004 (IV.21.) standard addressing the production and marketing of seed for agricultural crop species in Hungary, which was established following the International Seed Testing Association (ISTA). However, the ‘conditions necessary’ were altered to address our research questions.

### 1.2. Novelty

Overall, the novelty of this experiment of barley seed germination and seedling growth under abiotic stress lies in the unique combinations of experimental conditions, the innovative techniques and approaches used to measure plant responses, and the novel findings and insights generated by this experiment, which can contribute to our understanding of how seeds, seedlings, and plants cope with environmental stress and help in developing stress-tolerant crops.

This research establishes essential knowledge about germination requirements and investigates tolerance for various abiotic stresses.

## 2. Materials and Methods

This research examines several abiotic stresses, including temperature, water availability, seedling density, and antifungal influences on barley seeds’ germination vigor and development of the grown seedlings. Furthermore, the infection of fungal growth was investigated. The seeds of an extensively locally cultivated variety of Hungarian winter barley (Initium) were procured from a producer. The most significant benefit of the Initium is its drought tolerance, with a productivity of 8.0–9.0 t/ha. Its characterizations are as follows: 6-row head type, TKW is 42–49 g, hectoliter mass is 66–70 kg, and crude protein content of 12.3–13.9%. In addition, it has a good resistance level to powdery mildew, reticulate leaf spot, leaf rust, and *Rhynchosporium secalis*. The germination capacity can be 100%, which decreases with several factors, i.e., storage time and conditions, but it was more than 98% for the used stock. This research was undertaken at the Agronomy Institute/the Hungarian University of Agriculture and Life Sciences, Gödöllő, Hungary. Extremely sensitive and precise incubators manufactured by Memmert (HPP 260), Schwabach, Germany, with naturalistic convection or forced air influx and twin doors (internal glass and external stainless steel), were utilized to provide a clear vision without a temperature change.

### 2.1. Temperature Experiment

This research section was designed to investigate barley seeds’ germination ability at six variant temperatures (5, 10, 15, 20, 25, 30, and 35 °C). The petri dishes were marked, 20 barley seeds were situated in each, and similar quantities of distilled water were applied (7 mL) after conducting a conductivity test (1.5 umhos/cm). The experimental measurements include the ratio of germinated barley seeds, the shoots (the first proper leaf), and radicle lengths. The physical measurement began when approximately 70% of the seedlings reached a length of around 1 cm. Each day, four petri dishes were taken from the conditioned incubators of each temperature setting, and the seedlings were physically measured.

### 2.2. Water Availability Experiment

In sterilized petri dishes with a diameter of 9 cm and a single layer of sterilized filter paper, barley seeds were exposed to 13 different distilled water levels scaled at 1 milliliter intervals and 17 levels scaled in relation to TKW, as shown in Table 1. The TKW was acquired to be 38.33 g utilizing a seed counter instrument and an electronic scale.
[(TKW × Seed n)/100,000] = 1% of the suggested level of water(1)
[(38.33 × 20)/100,000] = 766.6/100,000 = 0.007666

The outcome of the above equation represents one percent of the proposed water level to be applied [11]. Therefore, it was multiplied by the proposed percentage for each treatment, as presented in Table 1. The petri dishes were marked and 20 seeds of barley were placed in each. They were implied at a constant conditioned chamber temperature of 20 °C with five replicates. Ten days after the settlement of the experiment, they were subjected to the physical measurement of measuring the length of the radicle and the shoot of all seedlings. Afterward, the radicles and the shoot separately were kept in a dryer at 65 °C for 2 days to obtain the dry weight of each treatment replication.

### 2.3. Seed and Seedling Density Experiment

This sub-experiment section was conducted to determine the effects of seed and seedling densities on germination and seedling performance at similar water levels in a petri dish. Twelve replicates (to increase the experimental precision) of each seed set, 10, 15, 20, and 25 per PD, were prepared for germination and then incubated in a chamber at 20 °C. These petri dishes were given a similar water quantity of 7 mL. Ten days after, they were subjected to the physical measurement of the radicle and shoot lengths. The acquired data were localized into five categories: not-germinated seeds (inactive seeds), initiated germination seeds, radicle-bearing only seedlings, seedlings with a short shoot (less than relatively 4 cm), and regular shoot seedlings. The phrase “regular seedling” refers to the morphological stage of a seedling compared to another in the same condition: a seedling with shoots more than 4 cm in length. 

These classifications were created in order to achieve an aggregated value following the Equation (2) [11]:Agg value = [(I * 0) + (IN * 0.1) + (R * 0.33) + (SS * 0.67) + (RS * 1)]/n(2)
where Agg value is the aggregated value, I is the number of inactive seeds, IN is the initiated germination seeds, R is the radicle-bearing only seedlings, SS is the number of short shoot seedlings, and RS is the number of normal shoot seedlings.

### 2.4. Antifungal Experiment

A fungicide, Bordóilé+kén NEO SC (hereafter referred to as Bordóilé), in varying concentrations using two varying application techniques, and Hypo were used to examine their impacts on preventing fungal growth in vitro. The active agents for this fungicide, Bordeaux juice + sulfur NEO SC 500 mL, are 215 g/L Bordeaux mixture + 290 g/L Sulphur. One technique was to apply 6 different concentrations of the Bordóilé fungicide, 0, 0.5, 5, 50, 500, and 50,000 ppm, to the growth media. The second technique was to separately sterilize the seeds into the prepared solutions of 1000 ppm of Bordóilé or 2% of Hypo for 90 s; afterward, they were rinsed with distilled water. The petri dishes were taped with parafilm tape to prevent water loss and contamination. Ten replicates for each treatment were incubated in the growth chamber at 20 °C. After a 10 day incubation period, physical assessments and measures were undertaken. The incubated petri dishes, radicle length, and shoot length were measured for each barley seedling, and the germination percentage was recorded.

### 2.5. Statistical Analysis

The data were checked for normality using Kolmogorov–Smirnov and Shapiro–Wilk in SPSS v27, IBM, New York, NY, USA. The results and presented data are expressed as a mean value. Analysis of variance (ANOVA) and Fisher’s test of least significant differences (LSD) as a post hoc test were utilized to indicate significant variations at the 5% probability level for the sub-experiments of temperature, water availability, seeds and seedling density, and antifungal using computing programs (GenStat twelfth edition, GenStat Procedure Library Release PL20.1m, and MS Excel 365). A sigmoid curve model was applied using statistical computing programs (J.M.P. Pro 13.2.1 of S.A.S. by SAS Institute, Canberra, New York, NY, USA, and MS Excel 365) to fit the data and plot the best-fit temperature levels.

## 3. Results

According to the statistics output (Table A1 (Appendix A)), as the Kolmogorov–Smirnov and Shapiro–Wilk are greater than 0.05, it can be declared that the curve is approximately symmetric [65,66], as illustrated by the histogram in Figure A1.

### 3.1. Temperature Experiment

#### 3.1.1. Germination Percentage

Data revealed significant variations in the germination percentages of barley seeds exposed to different thermal stresses. The temperature remarkably impacted the seed’s ability to germinate, as presented in Figure 1. A reversed relation exists between germination ability and temperature within the tested range as it gradually rises from 5 °C to 35 °C. At lower temperature levels, from 5 °C to less than 15 °C, germination percentages are maximum. Seeds can tolerate temperate thermal stress to germinate with a significantly lower percentage at more than a 20 °C to 30 °C temperature range. However, at 35 °C, barley seeds lose the ability to activate the germination process and are inhibited.

#### 3.1.2. Germination Duration

The gradient temperature experiment was designed to examine the ability of barley seed to germinate under stress, and the germination performance and pattern were reported as the time required to activate intracellular processes and enzymes and commence germination (Figure 2). The reference point of the duration measurement was deemed to be when 70% of the germinated seeds attained a shoot length of 1 cm. The thermal levels of 20 °C, 25 °C, and 30 °C accelerate germination commencement within 48 h. At 15 °C, barley seeds demanded more time (4 days) to accumulate the necessary temperature units to trigger the enzymes, commence germination, and attain a shoot length of 1 cm. Figure 2 demonstrates that barley seeds demanded more time to meet the measurement criteria (6 days) followed by those incubated at 5 °C with more extended time (10 days). The percentage differences between the two borders of germination ability of 5 °C and 30 °C is 133.333%, demonstrating significant disparities. Seeds incubated at a constant temperature of 35 °C could not germinate because they could not resist this stress level, and internal biological processes failed to be activated. There is marginal variation among germination duration with the temperatures of 20 °C, 25 °C, and 30 °C, followed by remarkable variations at 15 °C, 10 °C, and 5 °C, which required progressively more extended germination periods, in different steps.

#### 3.1.3. Seedling Development

Temperature is a determinantal factor affecting seedling growth rate and pattern [67]. The statistics of the temperature gradient study revealed that, at 20 °C, barley seedling growth development had the most remarkable performance and the most rapid growth rate compared to the other examined constant temperatures levels of 5, 10, 15, 25, 30, and 35 °C (Figure 3). It (20 °C) is the ideal temperature level for seedling growth and development with y-value = 8.8327x − 26.363 (Figure 2). Similar results, but with a slightly lower growth rate, were presented at a temperature level of 15 °C. Figure 3 presented that the seedling growth rate of barley at 10 °C demanded more time at early seedling stages but, in later stages, it was favorable to their development rate acceleration, with y-value = 7.8481x − 57.791 (Figure 2). Seedling growth patterns at 25 °C behave similarly to 20 °C, but with a slower rate even than that of 15 °C (Figure 3), with y = 7.559x − 21.778 as presented in Figure 2. A temperature level of 30 °C affected seedling development with a much slower growth rate than the lower temperatures (Figure 3), with y-value = 5.6072x − 17.534 (Figure 2). Figure 3 presented that seedling development at 5 °C is negligible and demands a much-prolonged development time, with y-value = 3.1399x − 33.145 (Figure 2). Barley seeds did not germinate at a constant temperature of 35; the seedling growth chart displayed zero values (Figure 3).

The shoots and radicles grew in the same pattern at and near the ideal thermal range, but their development performance varied as the temperature arrow crossed the ideal development range tails in either direction (Figure 4 and Figure 5). With temperatures surpassing the ideal range for barley seedlings’ growth and development, the shoots developed and expanded more rapidly than the radicles, predominately during the very advanced growth stage (Figure 4). Nonetheless, barley radicles appeared with a discretely different development pattern and required temperature than the shoots since the radicles developed better at a lower temperature than at the ideal range of the entire seedling development, especially in the later stage (Figure 5). The development and growth of barley shoots were uttermost when they grew at the temperature of 20 °C and the minimum at 5 °C, Figure 4. The optimal temperature range for radicle growth is 10–20 °C, Figure 5. The optimum temperature level for shoots and radicle development of barley seedlings was 20 °C, followed by 15 °C. There was no germination at 35 °C; the shoot and radicle growth charts showed them with zero values (Figure 4 and Figure 5). When the temperature level changes, either the shoot or the radicle will be more influenced than the other. While the radicle is more cold-resistant than the shoot, the shoot is more resistant to temperatures beyond the optimal threshold than the radicle.

### 3.2. Water Availability Experiment

Water absorption and temperature analysis can be used to identify the qualitative characteristics of barley seeds, such as their ability to withstand stress, uniformity, and germination rate [68,69]. *Hordeum vulgare* L. seed germination is severely constrained by moisture availability [70,71]. Thus, crop seeds with rigorous germination needs can be more effectively solicitously established than those of less restraint [13,72].

The current water availability experiment was conducted following two water application methods: 1 milliliter intervals starting from 0 to 12 mL, and water application correlated to TKW as percentages (17 water levels). It aimed to establish a new technique for comparing diversified barley varieties with assorted TKW in future studies. In addition, the two methods of water level application (a total of 30 water levels) were compared and concurrently evaluated to determine which provides a more accurate representation of water needs. The results demonstrate substantial differences in the water level potentials 0–12 mL for all the experimented traits: germination percentage, the length of seedlings, shoots, and radicles, the dry weight of the barley shoots, roots, and seedlings, and the adjusted dry weight of the seedlings that was achieved by eliminating the non-germinated seeds (Table 2).

Significantly higher germination percentages were observed as the water level grew up to the optimal water level, followed by a slight drop as the water level continued to increase owing to waterlogging (Table 2 and Table 3 and Figure 6). The two water application methods presented the same germination rate pattern with substantial disparities between the peak and the upper and lower tails because of the effect of drought and waterlogging stresses, respectively. The two-level polynomial curve of the germination percentages standardized in correlation to TKW as percentages in Figure 6, is smoother than the one standardized at one millimeter intervals. This may be an indication of the accuracy and reliability level of this technique. Water is fundamental for seed germination so their metabolic processes can be triggered at moistures adjacent to the critical threshold point. In the literature, cereal seeds begin germination when the volume of water absorbed reaches 40 percent of the seed’s size [73]. However, the metabolic processes of germination would not be activated when the interior moisture level is lower than the essential. Barley seeds could not germinate at 0.7 mL (Table 3), which exemplifies 90% of TKW (Table 1), and initiate germination at 1 mL. These results complement the study undertaken by [11,13,74], which concluded that seed size is significant for predicting germination under stressful circumstances. The optimum scaled range of water level for germination commencement is from 2.05 to 4.15 mL (Table 3), representing 270–540% of the TKW (Table 1). 

Water supply is crucial to seedling initiation development. The germination test, primarily conducted with vigor examinations such as a seedling growth test, is among the most outstanding seed quality and seedling performance tests [27,75]. There were substantial changes in seedling growth and development across different water level treatments. The average length of seedlings recorded the least values under conditions of insufficient water availability, and their performance increased dramatically as the water level increased (Table 2 and Table 3, Figure 7A,C). The ideal range of water availability begins at 5.5 mL to 8.25 mL, which, respectively, is equivalent to 720% and 1080% of TKW (Table 3). Simultaneously, considerably significant seedling development began at 4 mL according to the water supply at 1 milliliter intervals methods (Table 2). This suggests that TKW-derived percentages of water level supply are more consistent and accurate than those determined using constant intervals of water level application. Figure 7A,C illustrates that the ideal water level range for both water supplement methods is between 5 and 9 mL. It can be stated that moisture stress significantly diminishes seedlings’ vigor; nonetheless, physically, seedling vitality increases statistically substantially at the launching border of the ideal water availability range.

Significant variations among water levels were observed in the length of shoots and the radicles. As the water level rose, radicle length grew dramatically (Table 2 and Table 3). There is an ideal scaled range for the formation of barley radicles, which declines when the water level exceeds its top limit. The development model of the shoot is distinct from that of the radicle. Hence, evaluating the entire seedling length (Figure 7A,C) is reasonable. Drought and waterlogging biotic stressors affect the radicles more than the shoots of the barley seedlings.

The dry weight assay, an indicator of weight accumulation, is an essential measure of seedlings’ vigor and drought resilience. Dry matter accumulation varied greatly amongst the seedlings as a function of water level availability, as presented in Table 2 and Table 3. When the minimal level of water necessary for optimal development was surpassed, dry matter increased progressively (Figure 7A,C). Under stress, dry matter formation is depicted chronologically by amassing the necessary water level to produce one unit of dry matter. Therefore, drought and waterlogging reversely impact the formation of dry matter. Statistics show that shoot-accumulated dry matter needs more water than radicle-accumulated dry matter. According to the statistics, creating a unit of dry matter relating to the shoots requires more water than forming a unit of dry matter related to the radicles. This finding validates what was published by [11,74]. Water availability substantially influences the percentage of seeds germinating, seedling growth pace, and dry matter buildup. The decline in these parameters’ values is proportional to hydro stress intensity. Hydrological restraints and potential ranges exist for germination processes and the seedling growth progress phase.

### 3.3. Seed and Seedling Density Experiment

Statistics revealed no significant variations in germination percentages across various treatments of barley seeds and seedling densities of 10, 15, 20, and 25 per PD (Table 4). In addition, the aggregated values built up on the 5 subscales of life and time scale (Equation (2)), regarding seedling performance in different seedling densities, presented no significant differences among densities. Nevertheless, the subcategory of radicle-bearing only seedlings indicated significant differences with higher values linked to the central tested densities (15 and 20) and lower values linked to higher and lower ones. However, the remaining 4 subcategories—scilicet, the percentage of not germinated (inactive) seeds, initiated germination seeds portion, short shoot length seedlings portion, and regular shoot length seedlings portion—exhibited no significant differences. Accordingly, a density of 10 or 15 seeds per petri dish is preferable for the in vitro barley germination studies since a greater seedling density than the optimal had a contradictory impact upon opening the PD lid unless taping it.

### 3.4. Antifungal Experiment

This branch of this study describes the experiment undertaken to identify a technique to inhibit the development of fungi in petri dishes in vitro. The growth of the fungus is encouraged by higher temperatures. The parameters of an experiment change as the fungus grows, affecting the results. Data presented the remarkable impact of using the Bordóilé at a particular concentration in preventing fungal growth and giving healthy seedlings compared to the control (Figure 8A). The best concentration that prevented fungal growth and did not affect the germination and seedling growth when applied to the growth media was 5 ppm. The barley seeds grown in a media containing Bordóilé or Hypo presented a substantial favorable impact, stimulating germination and reducing fungal growth significantly compared to the control, with a greater positive value associated with the Hypo than the Bordóilé (Figure 8B). Two different techniques of antifungal application were tested: priming the seeds with fungi sterilizer solution and adding the antifungal directly to the growth media (Figure 8C). The seed priming technique was demonstrated to have a better effect on seedling growth and preventing fungal growth than applying the antifungal directly to the media.

## 4. Discussion

Abiotic stresses are non-biological factors that can adversely affect plant growth and development, including seed germination [11]. These stresses can include factors such as drought, extreme temperatures, and toxicity, among others [9,23]. While the effects of abiotic stresses on seed germination and seedling development have been widely studied, there is still ongoing research to uncover new insights and potential solutions to improve seed germination under stressful conditions. Overall, the ongoing research in abiotic stresses and seed germination aims to understand the underlying mechanisms of stress tolerance and develop new strategies to improve seed germination and crop productivity under adverse environmental conditions. In this experiment, they included specific abiotic stress conditions being tested with multiple levels of temperature, different bases of water application, different techniques of seed and media sterilization, different seed densities, the plant species being studied (barley), and the methods and techniques used to measure germination and growth, optimizing them and comparing them to previous studies.

### 4.1. Temperature Experiment

#### 4.1.1. Germination Percentage

The germination percentage is a remarkable estimate of the viability of a seed population, and it is crucial to crop establishment, yield, and quality [48,76]. To enhance agricultural output and quality, it is crucial to comprehend the germination characteristics of seeds under stress [77]. Temperature determines germination success [78,79]. The statistics demonstrated considerable differences in the germination rates of barley seedlings subjected to various heat stressors. Germination was greatly hindered by unfavorable temperatures, as demonstrated by Figure 1. Although seeds exposed to 35 °C did not germinate at all, a difference of 27.7372% between seeds sprouted between the maximum and lower limits of allowable germination temperatures, 5 °C to 30 °C. In comparison with previous studies of other crops, the percentage differences within the germination range between the upper and lower limits were 152.941% for wheat [13] and 177.778% for maize [11]. The maximum germination percentage was associated with 5 °C, indicating that this temperature level is ideal and slightly gradually lowers as the temperature rises to less than 15 °C, but with further increases, it declined dramatically. This indicates that greater temperatures alter the activity of enzymes and biological processes [11,77,80]. However, barley seeds can endure heat stress to germinate at a slightly reduced percentage at temperatures above 20 °C to 30 °C, but not higher.

#### 4.1.2. Germination Duration

Temperature significantly regulates germination duration [30,49,81]. The data demonstrate that the barley seeds accumulated heat units from the acquired thermal energy, and when they reached the required level to commence metabolic activity, germination occurred at varying rates depending on the ambient temperature [11,30,82]. According to the gradient temperature experiment, at temperatures between 20 and 30 °C germination commenced within two days of subjecting seeds to the treatment with no significant differences in reaching the same reference measurement point (Figure 2). However, to initiate germination, moderate temperatures around 20 °C are required. As the temperature rose over this threshold, the energy status and enzyme activity altered, causing a drastic decrease in protein synthesis and the ATP content rose dramatically [11,32,83]. Each germination stage requires a different temperature than the following [30,49]. In the very beginning, it needed a moderate to marginally higher temperature, and then it demanded a lower temperature since at 15 °C it needs more time to commence germination, but with higher germination percentages, and that is consistent with other research [13,80]. Barley seeds demanded a more prolonged time to meet the measurement criteria (6 days), followed by those incubated at 5 °C with more extended time (10 days). Significant disparities were presented between the shortest and the longest germination durations depending on the temperature level of stress, with a percentage of differences of 133.333%. Barley seeds failed to germinate at 35 °C, although they were monitored in the growth chamber for 30 days. In the scientific literature, one study indicated that the lowest germination temperature for barley seed is 4 °C and the maximin is 37 °C [84,85]. In this investigation, barley seeds could not germinate even at 37 °C, due to their exposure to a constant temperature. When the temperature level fluctuates day and night, 37 °C is barley’s maximum border germination range. Seeds can tolerate fluctuating temperatures near the upper germination range but not constant temperatures. Although 25 °C and 30 °C may swiftly commence germination, they have the opposite effect on metabolic activities throughout the subsequent germination phase. In similar conditions, barley seeds can better tolerate colder conditions than wheat, meaning that wheat requires a longer germination time starting from 20 °C and below [13]. In contrast, maize tolerated higher temperatures than barley and wheat, which required 48 h to germinate in a temperature range of 20–25 °C, but it could not germinate at 5 °C and needed a much more prolonged time under 10 °C and 15 °C [11]. 

#### 4.1.3. Seedling Development

Temperature is indispensable for crop seedling development, affecting all the different responses and growth stages [15,67]. Compared to the other evaluated constant temperatures, 20 °C yielded the highest performance and the uttermost seedling development rate (Figure 2 and Figure 3). The constant temperature of 20 °C followed by 15 °C is the ideal temperature range for barley seedling development, a finding which deviates slightly from those of other studies that declared the optimal range is 20–25 °C [86]. This is attributed to the accumulation of temperature in vitro at a constant temperature level, but the ideal range is somewhat greater with day-to-night temperature fluctuations. Seedlings grown at 15 °C followed a similar development pattern but with slower growth performance than those at 20 °C, which was also because of the accumulation of the necessary thermal units for each development stage. It demanded even more time for the same reason as seedling growth at a constant temperature of 10 °C. Even though a nearly similar growth pattern to that of the 5 °C was presented at 25 °C and 30 °C, nevertheless, it was far faster (Figure 3). This is due to the inverse influence of the high temperature levels on enzymes functions, protein synthesis, and increased biochemical energy content and ROS production, not because of the temperature accumulation [11,32,81,87,88]. At 25 and 30 °C, early seedlings exhibited a growth pattern that resembled one another to a small degree; they grew rapidly, then slowed down (Figure 3). Nevertheless, various seedling growth phases need different temperatures. This indicates that each stage of barley seedling growth requires a specific temperature threshold. The upper germination ranges of 25 and 30 °C resulted in rapid germination and a higher rate of seedling development very early, but not for the subsequent phases of seedling growth. Therefore, the ideal temperature range for seedling growth and development is 15–20 °C with a high growth rate as the temperature arrow goes toward 20 °C (Figure 3). When compared to previous studies, the seedling development of barley is slightly similar to wheat, but with more cold tolerance [13], and much different than maize [11].

The shoots and radicles grow in the same pattern at and near the ideal thermal range, but their development performance varied as the temperature arrow crossed the ideal development range tails in either direction (Figure 4 and Figure 5). With temperatures surpassing the ideal range for barley seedlings’ growth and development, the shoots developed and expand more rapidly than the radicles, predominately during the very advanced growth stage (Figure 4). Nonetheless, barley radicles appeared with a discretely different development pattern and required temperature than the shoots since the radicles developed better with a lower temperature than the ideal range of the entire seedling development, especially in the later stage (Figure 5). When the temperature level changes, either the shoot or the radicle will be more influenced than the other. While the radicle is more cold-resistant than the shoot, the shoot is more resistant to temperatures beyond the optimal threshold than the radicle. Barley roots tolerate cold conditions more than wheat. The shoots of the maize tolerate heat more than barley and wheat [11,13]. 

### 4.2. Water Availability Experiment

The germination assay is one of the primary measures of seed quality, and it is inextricably related to other seeds and seedlings’ vigor tests such as the seedling development test [89,90]. The hydrothermal duration of germination experiments may explain barley seeds’ tolerance to abiotic stress, uniformity, and germination viability. Water availability is potentially crucial to seed germination [53,91]. Cereal crops also have rigorous germination exigencies similar to barley [13,92]. Seeds with stringent germination criteria are more likely to sprout effectively than those with fewer constraints [30,73]. The water availability experiment was conducted following two water application methods: 1 milliliter intervals starting from 0 to 12 mL, and the base of water application correlated to TKW as percentages. This part of the study aimed to establish a new technique for comparing diversified barley varieties with assorted TKW in future studies. The proposed methods of water level application were compared and concurrently evaluated to determine which provides a more accurate representation of water needs. The results demonstrate substantial differences in the water level potentials for all the examined germination and seedling traits: germination percentage, the length of seedlings, shoots, and radicle, the dry weight of the barley shoots, roots, and seedlings, and the adjusted dry weight of the seedlings that was achieved by eliminating the non-germinated seeds (Table 2). These distinct impacts and diverse reactions of water availability on germination optimization are consistent with research conducted by [80].

The data indicate a distinct optimal moisture level for barley seed germination. As the water level rose to the optimal water level, germination percentages increased, then dropped less rapidly due to waterlogging (Table 2 and Table 3, and Figure 6). This is related to the water limit needed to trigger metabolic and physiological mechanisms to commence germination and the oxygen availability in more water levels. To begin, germination requires oxygen availability in sufficient amounts [93,94]. Since there is an inverse link between water and oxygen availability and an ideal balance for germination, seed oxygen availability dropped when water levels were applied over the optimum range. Therefore, the greater the water content, the less oxygen is accessible to the embryo and the biological germination process. Since moisture is a constitutional component for seed germination, germination can be commenced at moistures bordered to the critical point, which is enough to activate its processes.

The internal crucial moisture level prevents seed germination. Some cereals, wheat, for instance, demand an internal moisture level of 40% to commence the biological processes of germination [13]. Barley seeds cannot germinate at 0.70 mL (90% of the TKW), but they can commence germination with water application at 130% of the TKW. This availability of moisture level is adequate to trigger the metabolic processes of barley seed germination stages. The optimum scaled range of water level for germination commencement is from 2.05 to 4.15 mL (Table 3), representing 270–540% of the TKW (Table 1). Hence, the moisture application methods of the TKW offer a better understanding of the constraints and ideal moisture demands because seed size is essential for attaining the internal seed water level. These results are consistent with studies carried out by [80], indicating that seed size is significant for assessing germination under abiotic water stress conditions.

According to the data, remarkable variations in seedling growth and development across moisture levels existed. The lowest seedling length values are associated with low levels of water supplement, but they increased considerably with higher moisture levels (Table 2 and Table 3). The ideal range of water availability begins at 5.5 mL to 8.25 mL, which, respectively, is equivalent to 720% and 1080% of TKW (Table 3). Simultaneously, considerably significant seedling development began at 4 mL according to the water supply at 1 milliliter intervals method (Table 2). This suggests that TKW-derived percentages of water level supply are more accurate and consistent. Reduced enzymatic activity is among the numerous negative consequences of water stress on glucose metabolism, as are a lower water potential, lower levels of soluble calcium and potassium, and alterations to hormone levels of the seeds [95,96]. Since water stress influences these intracellular biological metabolisms, the seedling development eventually deflects from the average and declines.

Figure 7 demonstrates two distinct patterns of the shoots and the radicles of barley seedlings. The development of the barley radicle slowed progressively when the quantity of water exceeded the ideal range. Hence, it makes sense to gauge the whole seedling. There is a balance in their growth rate at the ideal range. However, the radicles are stimulated to elongate under drought stress, and the shoot growth declines. When the level of water availability increases more than the ideal range, radicle growth is hindered, the shoot’s performance and development increases to a tolerable limit and then slows under the high level of waterlogging, which is physiologically reasonable. The radicles were more susceptible to the adverse effects of water stress and waterlogging than the barley shoots. The formation-shoot-accumulated dry matter unit demands more water consumption than the generation of a unit of radicle dry matter. This conclusion is comparable with that of [74], in which researchers evaluated *Pinus yunnanensis* seed germination and seedling growth under different water availability and temperatures. According to Table 2 and Table 3, dry matter buildup of the seedlings varies significantly depending on moisture availability. The buildup of dry matter grows steadily until it reaches the ideal level, after which its pace accelerates. Under drought stress, dry matter formation is depicted chronologically by amassing the necessary quantity to produce one unit of dry matter. Barley seedlings growing in higher water levels than the ideal range are marginally negatively impacted by surplus water compared to the demanded level.

The severity of hydro stress substantially impacted the germination rate, seedling development pattern, and buildup matter; the more severe the hydro stress, the greater the drop in these parameters’ values. Hydrological restrictions and possible ranges exist for each stage of germination and seedling development [11,13,74,80].

In comparison with different crops under similar conditions, seed size is essential in stress conditions. Barley has a similar but broader range of the optimum water requirement based on seed size than wheat and maize [11,13]. This means barley can tolerate drought and water logging better than wheat or maize. 

In conclusion, this study suggests that TKW-derived percentages of water level supply are more accurate and consistent. Moreover, it supervises a better grasp of calculating the limits and optimal water needs since seed size is crucial to achieving the required internal seed moisture. Compared to other studies, these findings coincide with [11,74,80] where it was affirmed that germination rates and seedling performance could be evaluated more precisely when applying levels based on seed size, especially in studies when seeds are subjected to adverse environmental circumstances and stress.

### 4.3. Seed and Seedling Density Experiment

The findings shed light on which seed densities are necessary for successful seedling elongation bioassays in vitro. The level of phytotoxin present per seed and degree of inhibition increases as water volume exceeds the optimal upper level or as seedling densities of barley decrease [97,98]. In addition to internal and biological effects, there are physical effects of raising or reducing seedling densities, such as opening the lids of petri dishes, with a greater density than the ideal. This promotes vulnerability to water loss and, ultimately, stunts the development of the seedlings. Statistically, there were no significant differences in germination rates among the tested densities of 10, 15, 20, and 25 per PD, according to the data (Table 4).

Regarding seedling development, the aggregated value of assessing the collective seedling performance per petri dish as one unit revealed no significant differences across the investigated densities. Except for the radicle-bearing seedlings, when considered as a margin value, the remaining subcategories demonstrated no significant differences. Except for the radicle-bearing only seedlings, which gained only 10% of the aggregated value (Equation (2)), the remaining subcategories of the aggregated value demonstrated no significant differences. Seedling density can be optimized by avoiding high density, which has a reverse physical effect on opening the lid of a PD, and not low density, which may have a reverse statistical effect if 20% or more of the seeds fail to germinate. Therefore, seeds and seedling densities of 15 or 20 seeds of barley per a 9 cm petri dish are recommended for in vitro examination.

This result is consistent with previous studies on wheat and maize [11,13], where there were no significant differences among different densities in vitro. In the context of plant breeding programs (where identical seeds are frequently limited and optimization of the use of the resources is a crucial factor), determining the ideal amount of seeds to employ in a petri dish experiment is of critical relevance.

### 4.4. Antifungal Experiment

The proliferation of fungi harms seed germination and seedling development. Even in a laboratory setting, the development of fungi was shown to be temperature-dependent [99,100]. By adversely influencing the shoot and radicles of the growing seedling, this fungal development influences measurements of research parameters. This research was partly designed to identify a particular technique to inhibit fungal development in vitro. Data presented the remarkable impact of using the Bordóilé at a particular concentration in preventing fungal growth and providing healthy seedlings compared to the control. The best concentration that prevented fungal growth and did not affect the germination and seedling growth when applied to the growth media was 5 ppm. Although the antifungal, Bordóilé, inhibited fungal development, its high concentrations reversely impacted seedling growth. It impaired the cells’ osmotic pressure and the seeds’ water absorption. As the antifungal concentration in the growth medium increased more than the recommended limit, the germination percentages and seedling growth were significantly impacted. The application of Hypo presented the positive effect of stimulating germination and reducing fungal growth. The impact of the two procedures of priming with fungus sterilizer solution and applying antifungal, utilized to inhibit the development of fungi, was evident. According to the literature on antifungal actions on seed germination, it was declared that antifungal prevents and damages hyphae in vitro [99,101]. It damages DNA and proteins and lowers the fungus’ GSH level [11,101]. Therefore, antifungals inhibit barley seed germination issues and phytotoxicity on barley grains. Although the seed priming technique showed a remarkably much better impact on seedling growth than another technique with maize [11] in a previous study, the barley seed priming technique and the applying antifungal to the growth media technique, at an appropriate concentration of 5 ppm, were demonstrated to have a better effect on seedling growth and preventing fungal growth. This suggests that barley can tolerate a specific limit of antifungal application that is directly applied to the growth media. 

## 5. Conclusions

The sigmoid curves possess a solid fit for the experimental data representing seedling growth temperatures. The ideal temperature for barley seedling development is 20 °C followed by 15 °C and the range in between, and a more comprehensive range for germination rate is from 20 to 30 °C. A temperature lower than the optimal range decreases the germination rate (germination speed) but increases germination percentages, and a higher temperature raises fungal development.Seed size impacts the demanded level of water for germination. Hence, TKW administrates a more accurate base for water level application. Barley seed germination in different percentages can be initiated in a wide range of water levels starting at 1 mL, representing 130% of the barley TKW, but the optimal range for germination is 2.05 to 4.15 mL, representing 270–540% of the TKW. The optimal range for seedling growth is 5.5 mL to 8.25 mL, equivalent to 720% and 1080% of TKW.The dry weight measurement can be a reliable reference for seedling development since dry matter accumulation is compatible and consistent with the physical measurement of seedling development.There is no discernible difference in planting densities for seeds and seedlings in vitro; hence, planting densities as low as 10 barley seeds per 9 cm PD are advised for laboratory study.Both studied antifungal treatment techniques, seed priming and direct antifungal application at 5 ppm, effectively suppress fungal development. Seed priming is slightly superior to direct antifungal application on growth media, and both are significantly superior to the control.

This study proposes and recommends conducting barley seed germination and seedling development experiments at 15–20 °C, applying the water level as a percentage related to the TKW for water suitability optimization, and employing seed density of 10 seeds per PD, due to there being no significance in using a higher number of seeds. This is common practice in the case of seed limitation and breeding projects. Seed priming with antifungals, or antifungal application at 5 ppm of Bordóilé, significantly affected fungal growth inhabitation.

## Figures and Tables

**Figure 1 plants-12-01792-f001:**
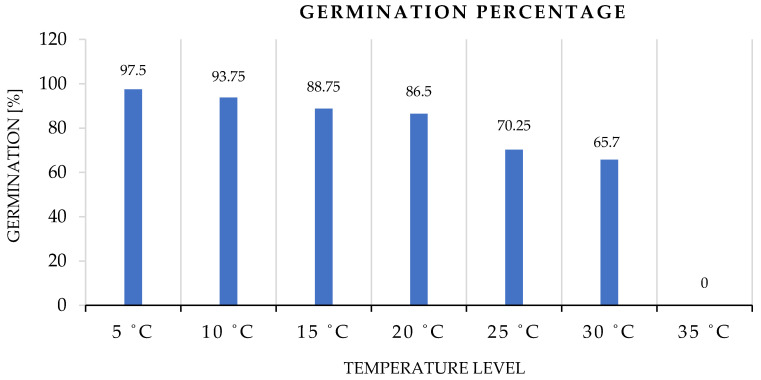
Germination ability of barley seeds with different levels of temperature treatment.

**Figure 2 plants-12-01792-f002:**
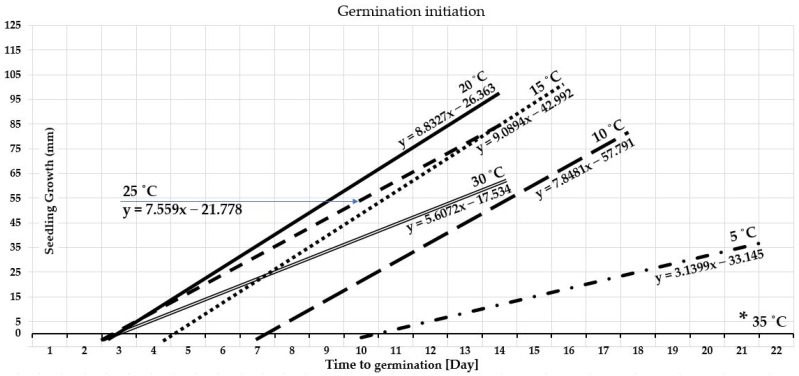
Timing of germination initiation under different levels of temperature stresses demonstrates the means of the seedling lengths for each day constantly after starting the measurement. The * indicates the 35 °C that is not demonstrated on the graph since its value is zero (no germination).

**Figure 3 plants-12-01792-f003:**
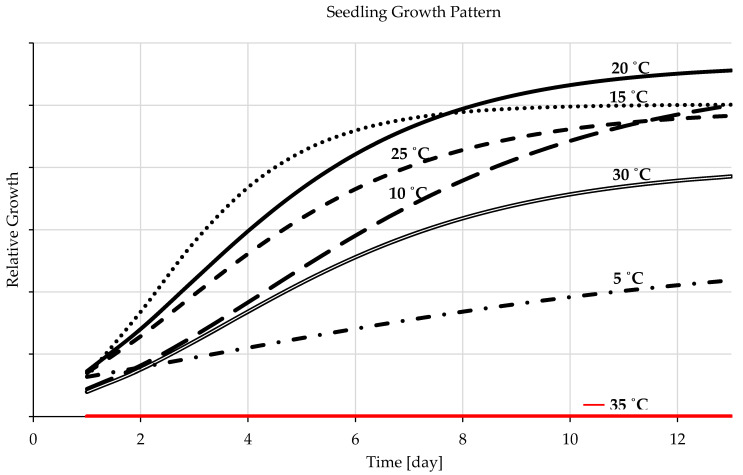
Temperature-dependent patterns of seedling development (day 0 is the germination initiation day).

**Figure 4 plants-12-01792-f004:**
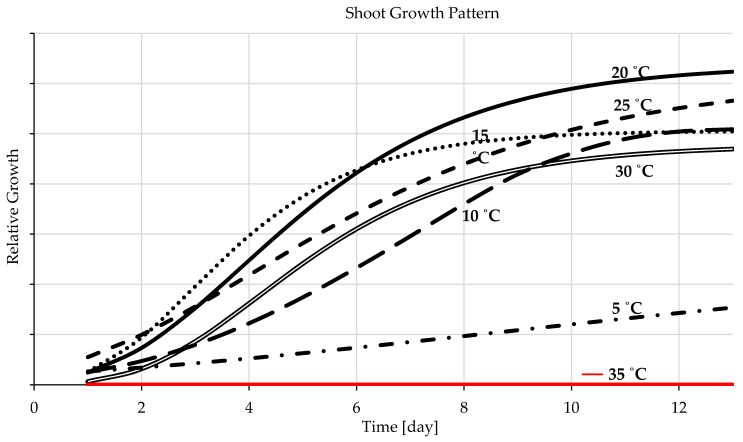
Temperature-dependent patterns of shoot development (day 0 is the germination initiation day).

**Figure 5 plants-12-01792-f005:**
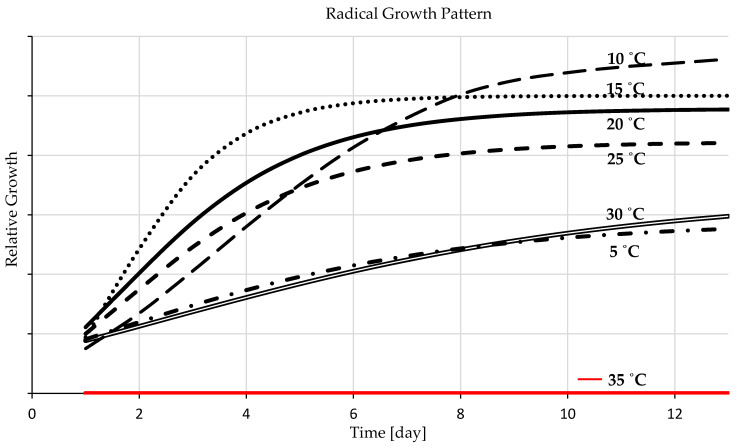
Temperature-dependent patterns of radicle development (day 0 is the germination initiation day).

**Figure 6 plants-12-01792-f006:**
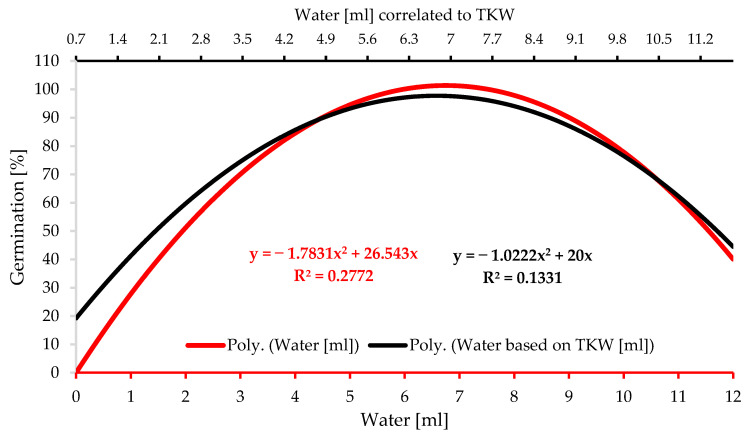
Barley seedling response to different levels of water availability standardized on 2 water application methods, as single-milliliter intervals and as a percentage of TKW.

**Figure 7 plants-12-01792-f007:**
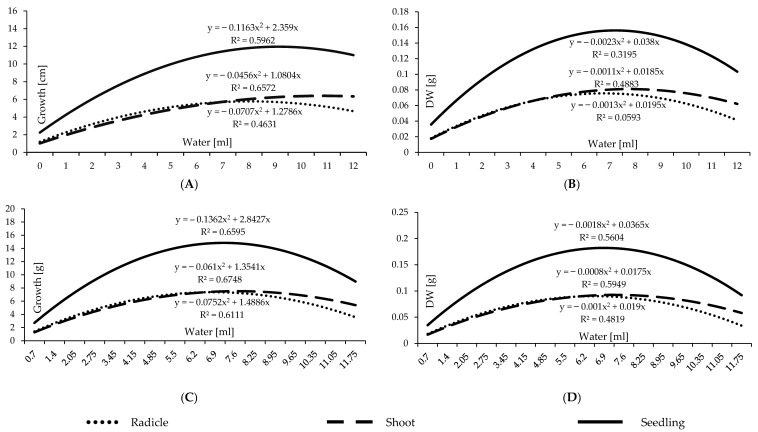
Barley’s radicles, shoots, and seedlings respond to the various water availability levels. (**A**) Growth reaction to water levels of 1 milliliter intervals (0–12); (**B**) dry weight accumulation as a reaction to water levels of 1 milliliter intervals (0–12); (**C**) growth reaction to various water levels supplied as a percentage correlated to the TKW; (**D**) dry weight accumulation as a reaction to various water levels supplied as a percentage correlated to the TKW.

**Figure 8 plants-12-01792-f008:**
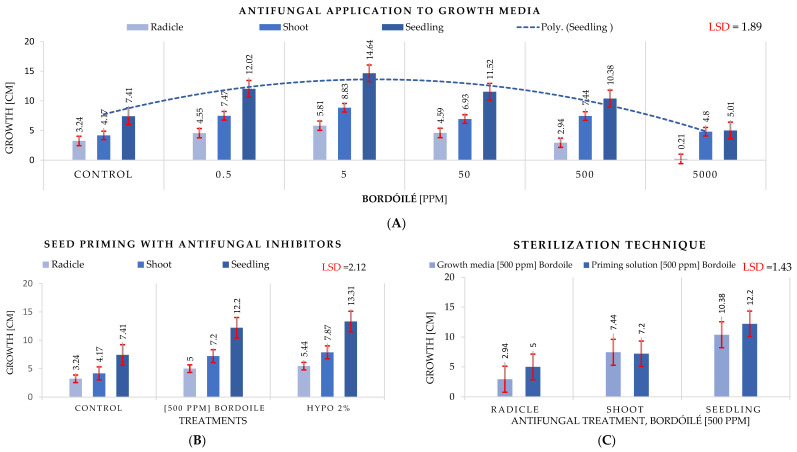
The effect of antifungal treatments and techniques on radicles, shoots, and seedlings of barley. (**A**) The effect of different concentrations of the applied antifungal (Bordóilé) to the growth media; (**B**) performance under antifungal and Hypo; (**C**) performance using two fungal growth inhibitor techniques (seeds were primed with sterilizer solution and antifungal media).

**Table 1 plants-12-01792-t001:** The experimental water quantity treatments scaled at one milliliter intervals and as a percentage of TKW.

Water Levels Scaled at 1 mL Intervals		Water Levels Scaled at the TKW %			
^1^ Treatment n^o^	^2^ Water Amount mL	^1^ Treatment n^o^	^3^ Proposed % of Water Level	^4^ level of Water mL	^5^ Rounded Level of Water mL
1	0	14	90%	0.69	0.7
2	1	15	180%	1.38	1.4
3	2	16	270%	2.07	2.05
4	3	17	360%	2.76	2.75
5	4	18	450%	3.45	3.45
6	5	19	540%	4.14	4.15
7	6	20	630%	4.83	4.85
8	7	21	720%	5.52	5.5
9	8	22	810%	6.21	6.2
10	9	23	900%	6.9	6.9
11	10	24	990%	7.59	7.6
12	11	25	1080%	8.28	8.25
13	12	26	1170%	8.97	8.95
		27	1260%	9.66	9.65
		28	1350%	10.35	10.35
		29	1440%	11.04	11.05
		30	1530%	11.73	11.75

^1^ The number of water treatments based on two water application scales; ^2^ the utilized water amount scaled at 1 mL intervals from 0 to 12 mL; ^3^ the suggested percentage of water level treatment concerning the TKW in g; ^4^ the level of water in ml concerning the proposed percentage of water level to TKW; ^5^ the rounded water level to the pipette’s next possible measurable digit (the actual used level in the experiment).

**Table 2 plants-12-01792-t002:** Germination and seedling measured traits of *Hordeum vulgare* L. seeds react to the water levels application standardized on a scale of 1 mL water intervals.

^1^ Water mL	^2^ Germination %	^3^ Radicle cm	^4^ Shoot cm	^5^ Seedling cm	^6^ Radicle DW g	^7^ Shoot DW g	^8^ Seedling DW g	^9^ Corrected DW g
**0**	0.0 ± 0.0 f	0.0 ± 0.0 d	0.0 ± 0.0 g	0.0 ± 0.0 e	0.000 ± 0.00 e	0.000 ± 0.00 d	0.000 ± 0.00 d	0.000 ± 0.00 c
**1**	90.0 ± 12.7 ab	3.0 ± 0.9 bc	0.7 ± 0.1 fg	3.7 ± 0.9 de	0.066 ± 0.02 abcd	0.023 ± 0.01 cd	0.090 ± 0.03 bc	0.099 ± 0.03 b
**2**	100.0 ± 0.0 a	4.7 ± 2.9 abc	3.1 ± 1.2 def	7.8 ± 4.0 bcd	0.094 ± 0.03 a	0.068 ± 0.03 abc	0.162 ± 0.05 ab	0.162 ± 0.05 ab
**3**	91.0 ± 5.5 ab	2.2 ± 1.2 cd	1.6 ± 1.4 efg	3.8 ± 2.6 de	0.038 ± 0.02 cd	0.077 ± 0.09 ab	0.115 ± 0.09 abc	0.130 ± 0.11 b
**4**	97.0 ± 2.7 a	6.8 ± 3.8 a	6.1 ± 3.1 abc	12.9 ± 6.9 ab	0.091 ± 0.05 ab	0.081 ± 0.04 ab	0.171 ± 0.09 a	0.177 ± 0.09 ab
**5**	84.0 ± 9.6 bc	4.9 ± 3.3 abc	5.8 ± 3.7 abcd	10.7 ± 7.0 abc	0.070 ± 0.04 abcd	0.080 ± 0.04 ab	0.150 ± 0.08 abc	0.174 ± 0.07 ab
**6**	79.0 ± 8.9 cd	5.4 ± 3.1 ab	5.9 ± 3.1 abc	11.3 ± 6.2 abc	0.063 ± 0.04 abcd	0.075 ± 0.05 ab	0.138 ± 0.09 abc	0.173 ± 0.11 ab
**7**	83.0 ± 12.0 bc	6.8 ± 2.8 a	7.3 ± 2.3 ab	14.1 ± 5.0 a	0.072 ± 0.04 abc	0.077 ± 0.03 ab	0.149 ± 0.07 abc	0.175 ± 0.06 ab
**8**	90.0 ± 7.9 ab	5.2 ± 2.6 abc	5.2 ± 2.7 abcd	10.4 ± 5.3 abc	0.059 ± 0.03 bcd	0.064 ± 0.04 abc	0.123 ± 0.07 abc	0.137 ± 0.07 b
**9**	77.0 ± 4.5 cd	2.9 ± 1.2 bcd	3.6 ± 2.2 cde	6.5 ± 3.3 cd	0.037 ± 0.01 d	0.037 ± 0.02 bcd	0.074 ± 0.04 cd	0.096 ± 0.05 b
**10**	66.0 ± 13.4 e	6.3 ± 1.8 a	7.9 ± 1.9 a	14.1 ± 3.6 a	0.067 ± 0.02 abcd	0.095 ± 0.03 a	0.162 ± 0.05 ab	0.244 ± 0.04 a
**11**	60.0 ± 6.1 e	4.1 ± 2.1 abc	4.8 ± 1.7 bcd	8.9 ± 3.8 abcd	0.047 ± 0.02 cd	0.061 ± 0.02 abc	0.109 ± 0.04 abc	0.178 ± 0.05 ab
**12**	69.0 ± 4.2 de	6.4 ± 0.7 a	0.0 ± 0.0 g	14.0 ± 1.3 a	0.070 ± 0.01 abcd	0.085 ± 0.01 ab	0.154 ± 0.01 ab	0.224 ± 0.02 a
**L.S.D ***	**10.17**	**2.94**	**2.74**	**5.58**	**0.035**	**0.049**	**0.077**	**0.084**

* Different lowercase letters [column] indicate statistically significant disparities in the means [*p* < 0.05], L.S.D.’s multiple commencing with the latter (a) is the most significant, ^1^ water treatment (mL) applied to a 9 cm PD, ^2^ germination percentage of the sprouted seeds proportionate to the total number per treatment, ^3^ the average length of radicles for the different treatments (cm), ^4^ the average length of shoot for the different treatments (cm), ^5^ the average length of seedlings for the different treatments (cm), ^6^ the dry weight averages of the radicles (g), ^7^ the dry weight averages of the shoots (g), ^8^ the dry weight averages of the seedlings (g), ^9^ the mean of the adjusted dry weight, which excludes the non-germinated seedlings.

**Table 3 plants-12-01792-t003:** Germination and seedling measured traits of *Hordeum vulgare* L. seeds react to the water levels application in correlation to the TKW.

^1^ Wml	^2^ Wof TKW	^3^ Germi %	^4^ Radicle cm	^5^ Shoot cm	^6^ Seedling cm	^7^ Radicle DW g	^8^ Plumule DW g	^9^ Seedling DW g	^10^ Corrected DW g
**0.7**	90%	0 ± 0.0 e	0.0 ± 0.0 h	0.0 ± 0.0 i	0.0 ± 0.0 k	0.000 ± 0.000 g	0 ± 0.000 h	0 ± 0.000 i	0 ± 0.000 i
**1.4**	180%	74 ± 8.3 bc	3.1 ± 41.4 g	1.0 ± 0.6 hi	4.1 ± 2.5 jk	0.051 ± 0.029 def	0.021 ± 0.012 gh	0.073 ± 0.041 gh	0.079 ± 0.044 h
**2.05**	270%	94 ± 1.3 a	5.4 ± 6.5 cdefg	4.2 ± 0.7 fg	9.5 ± 2.6 efghi	0.082 ± 0.024 abcd	0.074 ± 0.020 cd	0.156 ± 0.044 bcde	0.166 ± 0.045 efg
**2.75**	3605	92 ± 0.5 ab	4.2 ± 2.7 efg	3.2 ± 2.1 gh	7.4 ± 5.2 hij	0.061 ± 0.042 bcde	0.042 ± 0.035 fg	0.104 ± 0.076 efgh	0.113 ± 0.085 gh
**3.45**	450%	92 ± 1.3 ab	6.5 ± 6.7 abcde	5.7 ± 1.9 cdef	12.2 ± 3.7 cdefg	0.088 ± 0.021 abc	0.076 ± 0.019 cd	0.164 ± 0.039 abcd	0.179 ± 0.041 def
**4.15**	540%	89 ± 1.5 ab	7.2 ± 7.4 abc	6.4 ± 1.4 bcdef	13.6 ± 4.4 bcde	0.093 ± 0.036 a	0.081 ± 0.018 cd	0.174 ± 0.054 abcd	0.196 ± 0.060 bcde
**4.85**	630%	79 ± 1.8 abc	3.9 ± 8.9 fg	4.3 ± 2.3 efg	8.2 ± 4.4 ghij	0.048 ± 0.023 ef	0.045 ± 0.025 efg	0.092 ± 0.047 fgh	0.123 ± 0.074 fgh
**5.5**	720%	85 ± 2.1 abc	6.8 ± 10.6 abcd	6.6 ± 2.0 bcd	13.4 ± 4.2 bcdef	0.081 ± 0.023 abcd	0.080 ± 0.027 cd	0.161 ± 0.049 abcd	0.187 ± 0.048 cde
**6.2**	810%	84 ± 0.8 abc	9.0 ± 4.2 a	9.2 ± 1.0 a	18.2 ± 2.2 a	0.099 ± 0.007 a	0.109 ± 0.008 ab	0.207 ± 0.015 ab	0.247 ± 0.015 abc
**6.9**	900%	84 ± 3.3 abc	6.6± 16.4 abcde	7.7 ± 1.2 abc	14.3 ± 1.8 abcd	0.074± 0.010 abcde	0.091± 0.011 bcd	0.165 ± 0.018 abcd	0.201 ± 0.030 bcde
**7.6**	990%	81 ± 3.6 abc	7.9 ± 17.8 ab	8.5 ± 1.4 ab	16.4 ± 3.4 abc	0.082 ± 0.027 abcd	0.096 ± 0.015 abc	0.177 ± 0.041 abc	0.223± 0.044 abcde
**8.25**	1080%	80 ± 2.9 abc	7.3 ± 14.6 abc	8.4 ± 1.5 ab	15.7 ± 3.2 abcd	0.090 ± 0.021 ab	0.111 ± 0.020 ab	0.201 ± 0.038 ab	0.255 ± 0.046 ab
**8.95**	1170%	78 ± 1.1 abc	4.3 ± 5.7 defg	4.6 ± 1.8 defg	8.9 ± 3.0 fghi	0.059 ± 0.018 cde	0.070 ± 0.023 cde	0.129 ± 0.040 cdef	0.165 ± 0.049 efg
**9.65**	1260%	75 ± 4.1 abc	5.7 ± 20.6 bcdef	6.6 ± 3.1 bcde	12.2 ± 5.4 cdefg	0.060 ± 0.025 bcde	0.065 ± 0.027 def	0.125 ± 0.044 cdefg	0.165 ± 0.024 efg
**10.35**	1350%	82 ± 2.7 abc	8.2 ± 13.5 ab	9.8 ± 2.4 a	18.0 ± 5.3 ab	0.094 ± 0.037 a	0.121 ± 0.028 a	0.215 ± 0.065 a	0.263 ± 0.072 a
**11.05**	1440%	52 ± 3.1 d	4.5 ± 15.7 defg	6.9 ± 2.8 bc	11.4± 3.9 defgh	0.052 ± 0.023 def	0.069 ± 0.029 def	0.121 ± 0.051 defg	0.230 ± 0.054 abcd
**11.75**	15,305	69 ± 4.4 cd	2.8 ± 22.2 g	3.0 ± 0.9 gh	5.9 ± 1.5 ij	0.026 ± 0.004 fg	0.035 ± 0.015 g	0.061 ± 0.019 h	0.095 ± 0.033 h
**L.S.D ***		**19.96**	**2.57**	**2.25**	**4.59**	**0.031**	**0.027**	**0.055**	**0.062**

* Different lowercase letters [column] indicate statistically significant disparities in the means [*p* < 0.05], L.S.D.’s multiple commencing with the latter (a) is the most significant, ^1^ water treatment (mL) applied to a 9 cm PD, ^2^ water treatment level as a percentage correlated the TKW, ^3^ germination percentage of the sprouted seeds proportionate to the total number per treatment, ^4^ the average length of radicles for the different treatments (cm), ^5^ the average length of shoot for the different treatments (cm), ^6^ the average length of seedlings for the different treatments (cm), ^7^ the dry weight averages of the radicles (g), ^8^ the dry weight averages of the shoots (g), ^9^ the dry weight averages of the seedlings (g), ^10^ the mean of the adjusted dry weight, which excludes the non-germinated seedlings.

**Table 4 plants-12-01792-t004:** Categorical characteristics of density-dependent seed germination and seedling development of barley (*Hordeum vulgare* L.) per petri dish.

^1^ Seeds n^o^	^2^ Inactive Seeds %	^3^ Initiated %	^4^ Radicle-Bearing Seedlings %	^5^ Short Shoot Seedling %	^6^ Regular Shoot Seedling %	^7^ Aggregated Value %
10	0.110 ± 0.057 a	0.010 ± 0.032 a	0.010 ± 0.032 b	0.110 ± 0.099 a	0.760 ± 0.040 a	0.838 ± 0.022 a
15	0.133 ± 0.118 a	0.013 ± 0.028 a	0.027 ± 0.034 ab	0.180 ± 0.109 a	0.660 ± 0.058 a	0.791 ± 0.042 a
20	0.090 ± 0.088 a	0.035 ± 0.047 a	0.055 ± 0.060 a	0.145 ± 0.314 a	0.675 ± 0.087 a	0.794 ± 0.038 a
25	0.048 ± 0.056 a	0.028 ± 0.060 a	0.008 ± 0.017 b	0.248 ± 0.239 a	0.608 ± 0.117 a	0.780 ± 0.076 a
**L.S.D**	0.075 ^N.S^	0.040 ^N.S^	0.035	0.191 ^N.S^	0.232 ^N.S^	0.140 ^N.S^

^N.S^ refers to non-significant differences in [column] among the values of the means [*p* < 0.05], in accordance with L.S.D values, ^1^ the proportion of seeds per treatment, ^2^ inactive seed proportion of total seeds evaluated for each treatment, ^3^ seeds that began initiation of germination, ^4^ proportion of only radicle-bearing seedling, ^5^ proportion of short shoot seedlings (less than relatively 4 cm), ^6^ proportion of regular shoot length seedlings, ^7^ the aggregated value based on Equation (2) of the sub-classed fife groups: number of inactive seeds, seeds that initiated germination, only radicle-bearing seedling, short shoot seedlings, and regular shoot length seedlings.

## Data Availability

All data, tables, and figures in this manuscript are original.

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
