# Peer review of "Effect of Abiotic Stresses from Drought, Temperature, and Density on Germination and Seedling Growth of Barley (Hordeum vulgare L.)"

_plants, 2023, doi:10.3390/plants12091792_

Round 1
Reviewer 1 Report
The authors set a series of experiments aimed to scrutinize the germination capacity of barley seeds with different stresses conditions. As a conclusion, this study proposes and recommends conducting barley seed germination and seedling development experiments at 15-20 °C, applying the water level as a percentage related to the TKW for water suitability optimization, and employing seed density of 10 seeds per PD, and so on. Compared to other studies, this article designed different abiotic stresses and drew a scientific conclusion. There are still some problems in the article. Some suggestions for modification are as follows:
1. The topic of the article was “Abiotic Stresses of …”, while in the experiment design and the result, fungal infection condition was one of the content which was biotic stress.
2. In temperature experiment, 6 temperature levels were set. Why the temperature was form 5 to 30? Please explain the reason for this setting.
3. In seedling density experiment, the different petri dishes were given a similar water quantity. Whether the different height of water would affect the generation of seedling, please explain.
4. Figure 1 demonstrated the germination ability of barley seeds with different levels of temperature stresses; please describe the detail of the germination time, and the relationship between Figure 1 and another Figure of temperature-dependent experiment?
5. In conclusion, the conclusion and recommends should be converted to ha instead of per PD.
Author Response
Dear Professor,
Please find our answer in the attachment!
Respectfully,
The Authors

Reviewer 2 Report
Comments and Suggestions for Authors
Dear Author,
It is my pleasure to review the manuscript entitled “Abiotic Stresses of Drought, Temperature, Density on Germination and Seedling Growth of Barley (Hordeum vulgare L.)” a research article submitted to MDPI Journal, Plants. Authors of this manuscript investigate barley seeds' germination and seedling growth morphology under various abiotic stress conditions like temperatures, drought and waterlogging and identified their significant impact. Overall, the experiments are performed well and the results are convincing. Thus, the presented results take up an important topic consistent with the profile of the Journal. However, even, manuscript is well organized and well described of the conception, I have some suggestions, which might improve the manuscript to make important to the wider audience.
-This article lacks firm aim of the study that should be underlined precisely and simultaneously and highlight why this analysis is important to study.
-Some major and minor comments are as below
-There are many places where grammar can be improved. I suggest a careful revision by a professional language editor.
I've just noted a few here.
-Few suggestions I have mentioned in the main text pdf file. Please check
Title:
-Title should be specific with clear information.
Suggestion:
1. Effect of Abiotic Stresses from Drought, Temperature, and Density on Germination and Seedling Growth of Barley (Hordeum vulgare L.)
2. Seed Germination and Seedling Growth are severely affected by Abiotic Stresses from Drought, Temperature, and Density in Barley (Hordeum vulgare L.)
Abstract: -Good organization with results order.
However, need improvement in English and science. There is no line number, so it is difficult to comment.
1. Constant temperature levels of 5, 10, 15, 20, 25, 30, and were used. Drought and waterlogging stresses using 30 water levels were examined using two methods: one-milliliter intervals and as percentages of thousand kernel weight 17 (TKW).
-This sentence needs to be cleared with specific information
2. -What is two methods: one-milliliter intervals and as percentages of thousand kernel weight 17 (TKW)-indicate here?
3. Temperature significantly impacted germination time and seedling development with an ideal range of 15-20 °C, with a more comprehensive range to 10 °C.
-This lacks novelty. extreme low and high temperature is stressful for all living beings. The optimum range may be identified long time ago. Also sentence is not clearly informative.
4. Higher temperatures reversely affect germination percentage, and the lower ones affect the germination and seedling growth rate.
-35 °C is not too high temperature, therefore, did not affect seedling growth rate
5. Germination commenced at 130% water of the TKW, and the ideal water range for seedling development was greater and more extensive than the range for germination.
- better to use actual finding than vague explanation
6. The techniques of priming seeds with an antifungal solution (Bordóilé or Hypo) or antifungal application at 5 ppm significantly prevented fungal growth.
-What is result here? Antifungal agent obviously prevent fungal growth
7. This study is novel regarding the levels and types of abiotic stresses, the crop, the experimental and measurement techniques, and the comparison to the previous studies
-I think it is no necessary in abstract?
-Keywords: Need revision with important words. Barley, Hordeum vulgare L; may be redundant
-Also, better use alphabetic order
Comment in Introduction:
Introduction is not well organized and consistent with research goal. Most are general discussion rather specific scientific evidences. Needed to be more specific and sequential including some more specific findings referencing recent publications. Moreover, there are many redundant sentences. Therefore, my suggestion is, rewrite the introduction with most relevant information reviewing related recent published articles. Rationale with own findings to be elucidated at the end for the wider reader.
2. Materials and Methods
1. This research examines several abiotic stresses, including temperature, water avail- 159 ability, seedling density, and--------------------------------
This section should get a sub heading. Here many unnecessary wordings, sentences have been used which are not necessary for scientific information. Material collection source is not authentic and who have performed materials quality test, not indicated.
-Why do you need so many description for incubators?
2. The study was divided into five experiments, each according to the procedures outlined in the sub- sections below------
-Why this sentence is necessary? It is thesis writing style. Just to increase page no. is not good style for article. You may remove this type of sentencing throughout the text
3. Two types of experiments have been done here. i. seedling growth parameter study, and ii. Antifungal experiment ! what is relation between them in this mode of experiment. It just increased the volume of article. You may remove section ii. Experiment and discussion throughout the text. You also can make another article using those data as well
3. Results
According to the statistics output, Table A1 (Appendix), as the Kolmogorov-Smirnov 254 and Shapiro-Wilk are greater than 0.05, it can be declared that the curve is approximately symmetric [65,66], as illustrated by the histogram, Figure A2.
-This style is not usually followed in the article.
Line 261: …………as presented in figure TG.
What is Fig. TG???
Fig 1. No statistical significance. No SE mentioned.
L268: Figure 1. Germination ability of barley seeds with different levels of temperature stresses.
-Figure 1. Germination ability of barley seeds with different levels of temperature treatment.
-It would better, if authors could use at least one variety as a control that has tolerance to stress .
-Germination period up to 22 days?? Is it so?? Too long time germination continued?
Fig. 1 and Fig. 2: 5˚C showed best performance in Fig.1, whereas, start time too long in Fig. 2., However, in Fig. 3. Germination started at early stage upon treatment.
Some kind of misleading and contradictory information may present here
- For one temperature experiments, used many parameters and presented in different 4 figs. It is better to use all in one fig or as tabulated form
No. fig. has statistical significance test parameter
My Recommendation about this article:
Revision needed.
Needs rewriting as journal article format. It is most likely as a thesis style
Author Response

(The authors gave the same response as above.)

Round 2
Reviewer 2 Report
Accept in present form